# Muscle Hypertrophy and Architectural Changes in Response to Eight-Week Neuromuscular Electrical Stimulation Training in Healthy Older People

**DOI:** 10.3390/life10090184

**Published:** 2020-09-08

**Authors:** Tereza Jandova, Marco V. Narici, Michal Steffl, Danilo Bondi, Moreno D’Amico, Dagmar Pavlu, Vittore Verratti, Stefania Fulle, Tiziana Pietrangelo

**Affiliations:** 1Department of Neuroscience, Imaging and Clinical Science, Università degli Studi G. d‘Annunzio Chieti e Pescara, 66100 Chieti, Italy; te.jandova@hotmail.com (T.J.); danilo.bondi@unich.it (D.B.); damicomoreno@gmail.com (M.D.); stefania.fulle@unich.it (S.F.); tiziana.pietrangelo@unich.it (T.P.); 2Faculty of Physical Education and Sport, Charles University, 16252 Prague, Czech Republic; steffl@ftvs.cuni.cz (M.S.); pavlu@ftvs.cuni.cz (D.P.); 3Interuniversitary Institute of Myology (IIM), 66100 Chieti, Italy; 4Neuromuscular Physiology Laboratory, Department of Biomedical Sciences, University of Padova, 35122 Padova, Italy; 5SMART Lab (Skeleton Movement Analysis & Advanced Rehabilitation Technologies) Bioengineering & Biomedicine Company Srl, 66020 Pescara, Italy; 6Department of Psychological, Health and Territorial Sciences, University “G. d’Annunzio” of Chieti-Pescara, 66100 Chieti, Italy; vittore.verratti@unich.it

**Keywords:** sarcopenia, neuromuscular electrical stimulation, muscle ultrasound

## Abstract

Loss of muscle mass of the lower limbs and of the spine extensors markedly impairs locomotor ability and spine stability in old age. In this study, we investigated whether 8 w of neuromuscular electrical stimulation (NMES) improves size and architecture of the lumbar multifidus (LM) and vastus lateralis (VL) along with locomotor ability in healthy older individuals. Eight volunteers (aged 65 ≥ years) performed NMES 3 times/week. Eight sex- and age-matched individuals served as controls. Functional tests (Timed Up and Go test (TUG) and Five Times Sit-to-Stand Test (FTSST)), VL muscle architecture (muscle thickness (MT), pennation angle (PA), and fiber length (FL)), along with VL cross-sectional area (CSA) and both sides of LM were measured before and after by ultrasound. By the end of the training period, MT and CSA of VL increased by 8.6% and 11.4%, respectively. No significant increases were observed in FL and PA. LM CSA increased by 5.6% (left) and 7.1% (right). Interestingly, all VL architectural parameters significantly decreased in the control group. The combined NMES had a large significant effect on TUG (r = 0.50, *p* = 0.046). These results extend previous findings on the hypertrophic effects of NMES training, suggesting to be a useful mean for combating age-related sarcopenia.

## 1. Introduction

By the age of 80, people lose about 30–40% of skeletal muscle fibers and, on average, about 40% of strength [1,2,3]. This age-related loss of muscle mass and strength, termed sarcopenia, is a major contributor to loss of mobility, independence, and morbidity in the elderly [4]. Current estimates have revealed that about a quarter to a half of people older than 80 years are frail or sarcopenic [4,5]. The biggest concern is specifically the decline in the postural muscles of the back and the lower extremities. Research has demonstrated that the cross-sectional area (CSA) of the quadriceps muscle group in older individuals is reduced by 25–35% compared to young [6,7], and a recent longitudinal study by Fortin et al. [8] showed that age has a significant effect on the CSA of paraspinal muscles too. The most affected muscles are specifically the vastus lateralis (VL), the biggest muscle of the quadriceps responsible for actions such as walking or rising from the chair, and the lumbar multifidus (LM), an essential muscle for preserving an erect posture and for rotating the spine. When Lexell et al. [9] in 1988 examined the CSA of autopsied VL, they found a 39% decrease in fiber number with age; they also demonstrated selective atrophy of type II fibers with preservation of type I fibers. On the other hand, the CSA of LM has been found to be affected by age [10], with histological evidence (limited to patients with lumbar disc herniation or lower back pain) demonstrating atrophy of both type 1 and type 2 fibers [11,12]. When considering muscle function (muscle strength depends on the number of sarcomeres in parallel [13], the loss of muscle CSA is of serious concern.

Neuromuscular electrical stimulation (NMES), the application of an electric current to muscles in order to trigger muscle contractions, has been long used as an alternative intervention to resistance training in order to improve or attenuate muscle mass and strength losses [14,15,16]. NMES has proven to be efficient across different populations, ranging from healthy adults [17] and athletes [18] to people with muscle weakness [19] who cannot perform traditional volitional training. The efficacy of NMES stems from the fact that it enhances muscle protein synthesis [20] and preferentially targets type II muscle fibers (the ones that are affected the most by aging), which is contrary to the physiological recruitment pattern [21]. Several studies directed at molecular pathways of muscle hypertrophy demonstrated that anabolic hormones and muscular activity drive the mechanism through activation of the phosphatidylinositol 3 kinase/serine-threonine kinase Akt pathway (PI3K/Akt) [22]. This signaling pathway stimulates muscle protein synthesis through the activation of the mammalian target of rapamycin (mTOR) while inhibiting atrophy by phosphorylating the forkhead transcription factors FOXO. Phosphorylated FOXO is inactive, thus reducing the expression of muscle ubiquitin ligases, blocking the proteolysis [22]. At the molecular level, NMES has been suggested to cause positive adaptations in muscle signaling, inducing the upregulation of insulin-like growth factor 1 (IGF-1) and the differentiation of satellite cells [23,24]. Furthermore, research in young subjects has revealed that under certain conditions, NMES training may also improve muscle oxidative capacity, which could potentially enhance endurance performance [25]. Given that all these factors contribute to age-related sarcopenia, NMES seems to be an effective adjunct intervention in combating age-related sarcopenia. Moreover, studies have found that NMES is able to activate both types of muscle fibers (randomly with the preferential target of type II fibers), at relatively low force levels (25% of maximal voluntary contraction) compared to voluntary exercise in the elderly [26,27], which is of important clinical relevance in sarcopenia. In the elderly, numerous studies have demonstrated that particularly strength and functional performance improves significantly after NMES training [14,24,28]. However, these effects are less pronounced in healthy, active people [25,29], which suggests that NMES training may be more useful for people who are sarcopenic, unable, or unwilling to participate in traditional volitional activities or training.

The use of muscle ultrasound (MU) has been long advocated as a potentially reliable non-invasive tool for the quantification of skeletal muscle mass across different populations [30,31,32,33,34,35,36]. It has been used to assess structural muscle parameters and their changes in both hypertrophy (induced by exercise) and atrophy (induced by disuse), as well as in aging [37,38,39,40]. However, the morphological adaptations, in response to NMES in the elderly population, have not been sufficiently investigated yet. Hence, this study aimed to examine muscle size and morphological adaptations of VL and LM after eight weeks of combined NMES training applied to the quadriceps and lumbar paraspinal muscles in healthy older people using muscle ultrasound. We hypothesized that changes in muscle size and morphology in response to 8 w of NMES training would be detectable by MU given the underlying molecular basis of increased protein synthesis after NMES. In addition, we hypothesized that this change would not be necessarily linked to improved functional performance in healthy older adults. To our knowledge, this study was the first to analyze muscle morphology of LM and VL in response to NMES training using MU in this population of elderly.

## 2. Results

Calculated intraclass correlation coefficient (ICC) indicated an excellent level of reliability for all of the measured parameters (ICC ranging from 0.96–0.99) (Table 1). Mean current intensities reached towards the end of the NMES training were 33 mA (mean range from 26 to 37 mA) and 42 mA (mean range from 33 to 51 mA) for quadriceps and lumbar multifidus, respectively.

Sixteen volunteers participated in our pilot study. The median age of the volunteers (n = 16; NMES = 8 and control group (CG) = 8) in the NMES group was 69.3 ± 3.2 years and in CG was 68.0 ± 2.3 years. The study volunteers in the CG were taller and heavier than those in the NMES; however, the difference was not statistically significant. All of the participants were not obese according to the published body mass index (BMI) guidelines (41), with a median BMI of 26.5 ± 3.8 and 27.9 ± 2 in NMES and CG, respectively. Descriptive statistics of the study volunteers are presented in Table 2.

There was no significant difference in any of the measured parameters between the two groups at the baseline. From the measured MU parameters and following the quadriceps NMES training, CSA of VL significantly increased (pre-post) by 11.3 ± 15.1% (from 10.7 ± 4.6 to 12.2 ± 4.3 cm^2^) and muscle thickness (MT) by 6.9 ± 6.9% (from 1.7 ± 0.4 to 1.9 ± 0.4 cm). Interestingly, CSA of VL significantly decreased by 1.4 ± 4.0% (from 12.8 ± 2.6 to 12.5 ± 2.6 cm^2^) and MT by 2.4 ± 2.7% (from 1.9 ± 0.2 to 1.8 ± 0.3 cm) in the CG group. Moreover, pennation angle (PA) also significantly decreased by 0.6 ± 2.9% (from 13.9 ± 3.2 to 13.6 ± 3.0) and FL by 1.1 ± 4.2% (from 7.7 ± 0.7 to 7.5 ± 0.7 cm) in the CG group, whereas no significant change was observed in FL and PA in the NMES group. The NMES training of quadriceps had a large significant effect according to Cohen’s classification on all the measured MU parameters including MT (r = 0.84, *p* = 0.001), CSA of VL (r = 0.68, *p* = 0.006), PA (r = 0.68, *p* = 0.006) and FL (r = 0.50, *p* = 0.046).

In terms of the NMES training applied to the lumbar region, there was a 6.4 ± 8.2% significant increase (from 9.8 ± 2.7 to 10.4 ± 2.3 cm^2^) in the CSA of left LM and 6.5 ± 9.8% significant increase (from 9.1 ± 2.1 to 9.2 ± 2.6 cm^2^) in the CSA of right LM after the 8 w training period. No significant decrease was observed in CSA of both left and right LM after the 8 w period in the CG group. The NMES training applied to the lumbar region had a large significant effect on CSA of both right (r = 0.71, *p* = 0.005) and left (r = 0.53, *p* = 0.0036) LM.

There were no statistically significant pre-post differences observed in the functional parameters in both groups. However, the combined NMES training had a large significant effect on Timed Up and Go test (TUG) (r = 0.50, *p* = 0.046).

Median pre-post parameters (in units specific for each parameter) with interquartile range (IQR) are presented for both groups in Table 3. Percentage (%) changes, statistical significance, and effect sizes are presented in Table 4.

## 3. Discussion

Although NMES has been already primarily employed and proved effective as a means of resistance training across different populations of elderly [20,41,42], the morphological adaptations in response to NMES have been scarcely investigated. This pilot study reports on significant gains in lumbar multifidus (LM) and vastus lateralis (VL) size, based on architectural parameters assessed by ultrasound, after 8 w of combined quadriceps and lumbar paraspinal muscles NMES training in healthy older adults. Moreover, we found a large significant effect of NMES on functional mobility based around static and dynamic balance. Interestingly, we also found a significant decrease in VL muscle size and muscle architectural parameters (pennation angle and fascicle length) in the control group after the 8 w period. Together, these findings suggest that NMES training may play a protective role in age-related loss of muscle mass.

Indeed, the main findings of this study are that significant gains in muscle size of fundamental locomotor and spine erector and rotator muscles were obtained in response to the 8 w NMES intervention. The observed gains in muscle size and thickness were 11.4% and 8.6% for the knee extensors (VL), and for the left and right lumbar multifidus, the increase in muscle size was 5.6% and 7.1%, respectively. It is well-established that muscle CSA is an important physiological parameter for the evaluation of muscle size or the effectiveness of various exercise programs [43]. The gold-standard techniques of evaluation of muscle CSA are magnetic resonance imaging (MRI) or computed tomography (CT) [44]. However, recent studies have demonstrated that ultrasound-measured CSA [45] and MT significantly correlates with the corresponding portion of muscle anatomical CSA (ACSA) measured by MRI (see ref. [43] for review). Moreover, Franchi et al. [46] showed that MT changes at 50% of VL belly length were associated with corresponding changes in muscle ACSA (measured by MRI) after resistance training (RT). Therefore, the increased MT and CSA in our study are indicative of muscle hypertrophy following the combined NMES training. Indeed, ultrasound can measure only a focused area of the muscle and does not take into consideration the complete morphological adaptations to that muscle or other muscle groups, which is an advantage of MRI and CT. However, since the assessment of muscle mass in older people represents a significant problem both in research and clinical practice [47,48], muscle ultrasound with its non-invasive, fast, and real-time nature might be an attractive ecological alternative addition to the current assessment tool-box. In fact, our observed 11.4% increase in CSA of VL corroborates the result of past studies in older people, where the reported gains after 8–12 w of the traditional resistance training (knee extensors trained 3 x week, three sets at around 80% of one-repetition maximum) range between 9 and 15%, as determined by computed tomography (CT), fiber CSA or dual-energy X-ray absorptiometry (DXA) [49,50,51]. Therefore, the structural changes observed in our study not only suggest that NMES may have the same potency as the traditional strength training in gaining muscle mass, but it also suggests that our observed gains, as measured by ultrasound, are real.

The observed slight discrepancy between the left and right CSA of LM may be because the left CSA of LM was slightly bigger than the right CSA of LM at baseline (Table 3). According to literature, bilateral LM muscles are not symmetrical in pathological conditions, including acute or chronic lower back pain or lumbar disc herniation, which can cause muscle atrophy [11,52,53]. A study by Maffiuletti [29] has shown that LM muscle in patients with lower back pain (LBP) demonstrates atrophy of type I and type II fibers along with a significant reduction of CSA [29]. Therefore, it may be as well possible that our subjects had a slight degree of asymmetry, considering the fact that 13 to 50% of older people suffer from lower back pain [53], so the NMES training perhaps had a bigger effect on the slightly atrophied side. In fact, research has already indicated that better results of NMES are obtained usually in the presence of an impaired muscle condition [29,54]. In addition, Coghlan et al. [55] have already demonstrated improvements in LM performance, as evidenced by ultrasound evaluation of activation during voluntary activity after NMES, which was associated with improvements in self-reported pain levels. This is a clinically interesting finding, suggesting that NMES of LM may be effective in targeting LBP in addition to improving muscle size. Improving LM mass likely benefits posture maintenance and spine rotation, which is essential for preserving performance in daily activities in older individuals. With this respect, future studies could consider employing techniques based on stereo-photogrammetry, such as the Global Optoelectronic Approach for Locomotion and Spine [37,56], in order to investigate the outcomes of such interventions on spine and postural parameters in the elderly.

While it is well-established that resistance training (RT) increases PA and, if involving eccentric contractions, also FL [57], only a small, not significant, increase in these parameters were found in this study. However, this does not seem surprising since, in our study, as in most NMES protocols, muscle contraction was evoked in isometric conditions. While an increase in muscle size necessarily involves the addition of sarcomeres in parallel, this should thus be reflected by an increase in pennation angle. In fact, an increase in PA is considered as a space-saving strategy, and a large number of studies already demonstrated an increase in PA with muscle hypertrophy following resistance training [58]. Instead, an increase in fascicle length, reflecting the addition of sarcomeres in series, is typically observed after eccentric exercise training [59,60]; thus, involving lengthening contractions. Indeed, Reeves et al. [60] showed on the same number of older volunteers who were randomly assigned to either eccentric-only RT or conventional RT that FL (along with MT) increased in both groups but PA only in the conventionally RT (by 35%). As a matter of fact, it is now well-recognized that the direction of muscle growth depends on the mode of muscle contraction [57], therefore, since our NMES protocol involved static rather than shortening and lengthening contractions, the tendency for an increase in FL (1.4%) and PA (4.4%), observed after NMES may be simply the result of the isometric training mode. Moreover, several studies have previously reported that the adaptation of both FL and PA may be unlikely to occur in the elderly, despite increases in MT and CSA [61,62]. For example, Noorkoiv et al. [63] demonstrated that isometric training at a long muscle length position (a knee joint fixed at a 90° knee extension) induced an increase in VL FL in the distal region, which may suggest for regional architectural differences, and partially explain the results of our, and the latter two studies [61,62], given the localized nature of ultrasound measurements.

The significant decrease in all the architectural parameters of VL (Table 3) in our CG group may be indicative of muscle disuse atrophy. In disuse atrophy, as in sarcopenia, the decrease in muscle mass is accompanied by a decrease in FL and PA [34]. In the elderly, sarcopenia is accelerated by physical inactivity [64], affecting primarily the lower extremities [65]. Therefore, we could speculate that our CG subjects may have reduced their level of physical activity, such as walking. In fact, our study was conducted during colder months, which may have affected the physical activity level of the volunteers. Nevertheless, these findings point out the importance of physical activity in the elderly to counterbalance age-related sarcopenia, and in the view of these findings, NMES seems to have a protective effect.

Considering that NMES is directly applied to the muscle and does not involve any volitional drive to the muscle, it could be argued that NMES-induced contractions require little activation of the motor cortex and are thus unlikely to improve functional performance. However, NMES, when applied to the peripheral muscles, seems to have a direct effect on the cerebral cortex since activation of the primary sensorimotor cortex and the supplementary motor area has been described in response to peripheral muscle stimulation in humans [66]. Therefore, it could be argued that the lack of pre-post improvements in functional performance was due to either asymmetry in muscular vs. neural gains, where NMES had a stronger effect on the latter or due to the very good physical performance of our elderly volunteers in both functional tests at the beginning of the study [67,68]. Nevertheless, the combined NMES had a large significant effect on TUG, which is a measure of functional mobility around both static and dynamic balance, which is in line with the findings of other studies, where improvements in functional performance in elderly individuals after NMES training were noted primarily for posture and balance parameters [14,23,24,28].

It was not the purpose of this study to measure the cellular and molecular responses that promote muscle growth such as the production of IGF-1, protein synthesis rate, gene transcription factors, or translational mechanisms. However, previous research has shown that these adaptive processes take place even after only one acute bout of NMES [69,70]. It has been well-demonstrated that NMES in healthy older people, increases the expression of IGF-1, myogenic regulatory factors (PAX3, PAX7, MYF5, MYOD, MYOG, N-CAM), micro-RNAs (miR-1, miR-133a/b, and miR-206) and activates satellite cells [23,24,71]. The expression of IGF-1 is typically up-regulated in response to the mechanical stimulus of muscle overload [72]; therefore, the detected hypertrophic structural changes in muscle architecture in the present study could be mostly ascribed to the cellular and molecular mechanisms behind muscle hypertrophy induced by NMES training. As a matter of fact, our NMES loading with intensities corresponding to the individual’s pain threshold has already proved effective in improving various muscle parameters in previous studies including both quadriceps and lumbar multifidus muscles [23,24,55]. Altogether, our results extend previous findings of cellular and molecular studies on the hypertrophic effects of NMES training in the elderly muscle [71,73,74,75] by the mean of the non-invasive method of muscle ultrasound.

It should be noted that some methodological considerations are involved in this study. First, the main limitation of this study is the small sample size. However, our specific department rules and guidelines do not allow bigger groups at one time to guarantee maximum safety and surveillance of recruited volunteers. For this reason, we used the non-parametric statistical testing as the recommended approach in medical and biomedical science research [76,77] along with the calculation of effect sizes to account for this problem. Future studies involving a larger cohort of subjects are needed to confirm the findings of this study further as well as to investigate the potential sex differences in the training response. Future studies should also ideally closely monitor the activity and behavior patterns of the elderly in control groups to understand observed changes in muscle morphology better. Second, we were not able to measure the force produced during NMES stimulation and express it as % maximum voluntary contraction (MVC) due to incurred technical problems. However, we can speculate based on the findings of Marmon and Snyder-Mackler [78] that the NMES intensities reached by our subjects may have been around 50% of MVC. In future studies, each participant should be perhaps ideally stimulated at the same % of some measurement of strength or function as the individual tolerance varies enormously for a variety of uncontrollable reasons [29]. Other methodological issues that should be mentioned are related to the actual technique of MU. For example, the FL of VL muscle is longer than the 5 cm width of the probe used in this study. Therefore, it is difficult to visualize the whole FL in an image. In our study, we used the linear extrapolation technique (Figure 1), and although validation of FL estimation using this method can be confirmed in VL [79], an actual underestimation of FL for example due to a three-dimensional fascicle arrangement [80] or fascicle curvature [81] may still be possible. However, the values of FL for the VL of our elderly individuals are in line with those reported in elderly populations by Reeves et al. [82].

In conclusion, the present pilot study demonstrates that hypertrophic changes in the lumbar multifidus and vastus lateralis muscles can be induced by an eight-week NMES intervention in older individuals and detected by muscle ultrasound. This study, therefore, confirms and extends previous findings on the hypertrophic effects of NMES in the elderly, introducing newly the NMES of paraspinal muscles to target lumbar multifidus. Interestingly, the results of this study also show a deterioration of VL muscle size in control non-training older individuals, likely attributable to sedentarism of this group. Taken together, these results suggest that NMES may have a protective effect in age-related loss of muscle mass. However, future studies with a larger number of subjects are needed to confirm these findings further.

## 4. Materials and Methods

### 4.1. Participants

Sixteen healthy older volunteers (males = 8, females = 8; aged ≥ 65 years) were recruited from the community for this study. All volunteers were considered healthy and suitable for the study after meeting the following exclusion criteria: no active cardiovascular, cerebrovascular, respiratory, metabolic, active inflammatory bowel or renal disease, malignancy, recent steroid treatment (within six months) or hormone replacement therapy, blood clotting dysfunction, musculoskeletal or neurological disorder, family history of early (<55 y) death from cardiovascular disease, muscular or osteoarticular injury in the previous six months, self-reported physical activity at moderate-intensity exercise (≥30 min/day, 3 × week) or regular participation in strength training. The study was approved by the University of “G. D’Annunzio” Chieti-Pescara Ethics Committee (doc. n.16 of 05/09/2019), conforming to the ethical standards set by the Declaration of Helsinki [83]. All volunteers provided their informed written consent to participate in the study.

### 4.2. Experimental Design

The sixteen volunteers were divided according to sex and then randomly assigned using free online statistical computing web programming (www.graphpad.com/quickcalcs) that generated the randomization schedule to two groups—the control group (CG) and electrostimulation group (NMES). Functional tests and muscle architecture of vastus lateralis (VL), including muscle thickness (MT), pennation angle (PA), and fiber length (FL), along with CSA of VL and both sides of the lumbar multifidus (LM) muscle were measured before and after the 8 w NMES training period using muscle ultrasound.

### 4.3. Combined NMES Training

Combined NMES consisted of three sessions per week for eight weeks. All volunteers performed a 5 min warm-up (at the beginning of each session) using a cycle ergometer (Bike 600XT PROTM, Technogym^®^, Gambettola, FC, Italy) at a light intensity (≤60% of their predicted maximal heart rate (HRmax)), calculated by the Tanaka formula [84] and monitored on the screen of the cycle ergometer. At the end of each session, all volunteers also performed a light 5 min lower back and quadriceps stretching exercise. During NMES sessions, the transcutaneous electrical stimulation was delivered through a set of four (5 × 5 cm) fully gelled electrodes with a contact area of 25 cm^2^ using an NMES device (Genesy 1200 Pro; Globus Srl, Codogne, Italy). NMES training of the quadriceps muscle group was set according to our previous laboratory protocol described by Di Filippo et al. [23]. NMES protocol of lumbar paraspinal muscles (targeting the LM) was experimentally set based on the previous work of Baek et al. [85], Kim et al. [86], and Coghlan et al. [55] due to demonstrated efficiency across different groups of patients including also lower back pain patients.

### 4.4. NMES of the Quadriceps Muscle Group

The NMES of the quadriceps muscle group was performed first. Volunteers were seated with a knee joint fixed at a 90° on a leg extension machine to allow for an isometric condition during stimulation. After cleaning the skin with an alcohol wipe, two active electrodes were placed over the motor points of the quadriceps muscles of vastus lateralis and vastus medialis identified by palpation, and two dispersive electrodes were placed approximately 5–7 cm below the inguinal crease to close the stimulation loop [23] (Figure 2A). Rectangular biphasic wave pulsed currents (75 Hz per 400 µs) were delivered with a rise time of 1.5 s, a steady tetanic stimulation time of 4 s, and a fall time of 0.75 s (total duration of the contraction: 6.25 s) with a rest interval of the 20 s between contractions in order to minimize discomfort, muscle fatigue, or muscle damage [87]. The NMES training lasted 18 min in total and consisted of 40 passive isometric bilateral contractions.

### 4.5. NMES Training of Lumbar Paraspinal Muscles

The NMES of quadriceps was followed by 15 min of NMES training of lumbar paraspinal muscles. Volunteers were seated in a relaxed position on an armchair. After cleaning the skin with an alcohol wipe, four electrodes were placed bilaterally, as in [85], at the vertebral levels of L4 and L5 located previously by iliac crest palpation [88] (Figure 2B). Symmetrical biphasic wave pulsed currents (50 Hz per 300 µs) were delivered with a ramp-up for 1 s, a steady tetanic stimulation time of 8 s, and a ramp down for 1 sec (total duration of the contraction: 10 s) with a rest interval of 10 s between contractions. The transcutaneous electrical nerve stimulation (TENS) setting was applied for the rest interval to suppress any possible pain signals for better relaxation of the muscles [55].

For both NMES training protocols, the intensity was monitored every 3–5 min and gradually increased to reach the maximum tolerable intensity during each session, corresponding to the individual’s pain threshold. Volunteers were also instructed to freely adjust the intensity during the training session to reach their maximal tolerable intensity without feeling discomforts, such as burning sensation or severe tetanic pain. The schedule of the combined NMES training detailing the duration, frequency, load, and intensity is described in Table 5.

### 4.6. Functional Tests

Five Times Sit-to-Stand Test (FTSST) and Timed Up and Go test (TUG) were carried out to assess the volunteer’s functional capacity. FTSST to assess the postural control and lower limb muscular strength and TUG the functional mobility of volunteers, based around both static and dynamic balance. FTSST was performed according to Wallmann et al. [89], whilst TUG was performed according to the original study by Richardson [90]. Both tests were repeated three times using a 43 cm highchair with approximately two min rest in between each trial. The best performance value was used for later statistical analysis.

### 4.7. The Procedure of Ultrasound Scanning

The same investigator performed all muscle ultrasound (MU) measures using a B-mode US (MyLab Gamma, Esaote Biomedica, Genova, Italy) equipped with a 5 cm, 3–11 MHz linear-array probe. For both protocols, seven volunteers were measured before the start of the training period in order to test for inter-day reliability by intraclass correlation coefficient (ICC) [36].

MU of VL followed the protocol by Ticinesi et al. [91]. Muscle architecture including fiber length (FL), pennation angle (PA), and muscle thickness (MT) were measured on the right site of resting VL at 65% distal distance by placing the MU probe longitudinally along the VL long axis. For the images of CSA of VL, the extended field of view (EFOV) software was activated with the probe was rotated 90° counterclockwise at the same 65% distal distance [91]. During the MU of VL, the study volunteer was lying supine on an examination bed with the knee fully extended (anatomical zero). Three longitudinal images of VL and three CSA images of VL were taken and saved for later analysis [91].

On the other hand, MU protocol of LM was based on the procedure of Stokes et al. [66,92]. Briefly, the study volunteer was relaxed in a lying prone position on an examination bed with one or two pillows placed under the hips to minimize lumbar lordosis. The CSA of resting LM was measured bilaterally at a vertebral level of L5, which was previously palpated by the iliac crest palpation [88] and marked by dermographic pen. MU was used to confirm and verify the skin marks by placing the probe longitudinally over the lower lumbar spine in the mid-line. After this, the probe was rotated 90° to lie transversely at L5 level in the midline to capture the CSA of LM. Three images of both sides of LM were taken and saved for later analysis [92].

### 4.8. Analysis of Ultrasound Images

The acquired ultrasound images were analyzed off-line using the free NIH Image-J software [93] by the same investigator. VL was analyzed according to the procedure of Ticinesi et al. [91], whereas LM was analyzed according to the procedure of Stokes et al. [66,92]. Figure 1 shows the measurements of VL and LM graphically. At least one measurement of all the MU parameters was obtained for each of the three images, and then the average was calculated and used for later statistical analysis.

### 4.9. Statistical Analysis

ICC—two-way random, absolute agreement, with 95% confidence intervals (CI) was calculated from two sets of ultrasound measurements performed on two separate days. Non-parametric methods were used as the recommended approach in medical and biomedical science research [71,72] given the small number of subjects. For descriptive statistics, medians, and interquartile range (IQR) were calculated. Between-group differences for the descriptive statistics and baseline values were tested by the Mann–Whitney U test. The non-parametric Wilcoxon signed-rank test was used to test the within-group difference before and after the intervention period for each group separately. The Mann–Whitney U test was then used to test the changes (pre-post) between the groups. For the effect size, we calculated a correlation coefficient r using Z value from the Mann–Whitney U test as r = Z/√N, where N is the total number of the participants. Standard interpretation for r is the same as according to Cohen’s classification of effect sizes where 0.1 = small effect, 0.3 = moderate effect, and ≥0.5 = large effect. All the statistics were carried out in IBM SPSS Statistics 24.

## Figures and Tables

**Figure 1 life-10-00184-f001:**
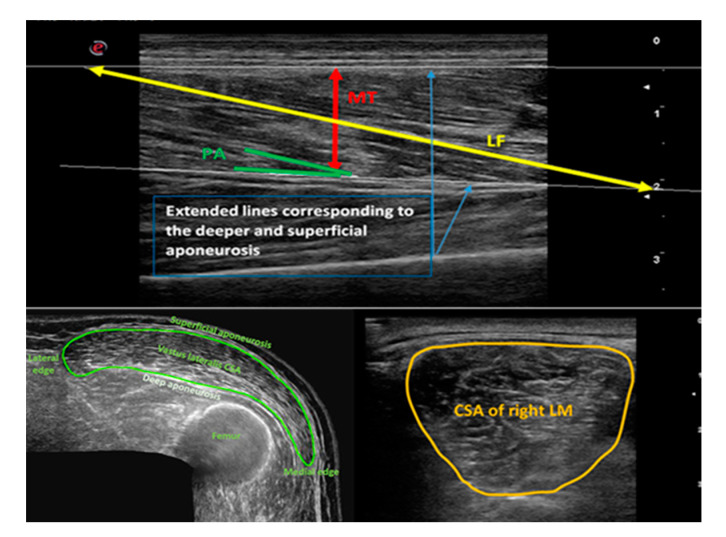
Graphical representation of vastus lateralis and lumbar multifidus ultrasound image analysis. Upper panel: Measurement of vastus lateralis (VL) architecture (MT = muscle thickness; LF = fiber length; PA = pennation angle); lower panel: measurement of cross-sectional area (CSA) of VL and lumbar multifidus (LM) as traced by a cursor around the muscle border.

**Figure 2 life-10-00184-f002:**
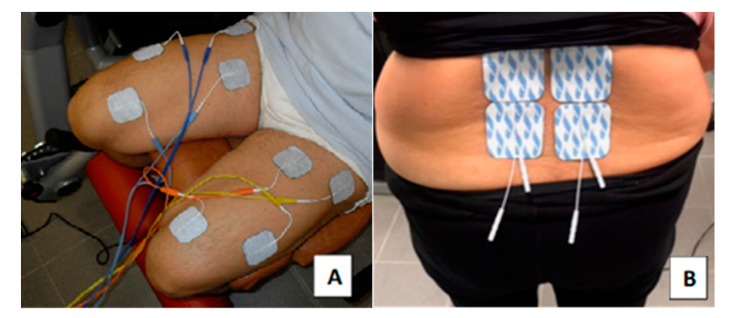
Combined neuromuscular electrical stimulation (NMES) set-up. The positioning of electrodes on the quadriceps muscles (**A**) and of the lumbar paraspinal muscles (**B**).

**Table 1 life-10-00184-t001:** Intraclass correlation coefficient for muscle ultrasound parameters.

	Intraclass Correlation Coefficient	95% CI	
		Lower Limit	Upper Limit
CSA of VL	0.988	0.849	0.998
MT of VL	0.994	0.948	0.999
FL of VL	0.971	0.752	0.995
PA of VL	0.948	0.577	0.990
CSA of LM left	0.970	0.854	0.994
CSA of LM right	0.994	0.974	0.999

Note: CI = Confidence interval; CSA—cross-sectional area; VL—vastus lateralis; MT—muscle thickness; FL—fiber length; PA—pennation angle; LM—lumbar multifidus.

**Table 2 life-10-00184-t002:** Descriptive statistics of the study volunteers.

	NMES *n* = 8	CG *n* = 8	*p* Value *
Age (y)	69.3 (3.2)	68.0 (2.3)	0.588
Weight (kg)	67.6 (15.9)	80.1 (10.7)	0.066
Height (cm)	160.1 (8.4)	170.4 (9.2)	0.058
BMI (kg/m^2^)	26.5 (3.8)	27.9 (2.0)	0.294

Note: values are presented as a percentage for sex distribution and medians (interquartile range, IQR) for the other variables; * significance was tested by chi-squared test (sex distribution) and by Mann–Whitney U test (the other variables); (BMI = body mass index, NMES = neuromuscular electrical stimulation group, CG = control group).

**Table 3 life-10-00184-t003:** Measured muscle ultrasound parameters at baseline and after 8 w of combined NMES training.

	NMES *n* = 8		CG *n* = 8	
	Pre-	Post-	Pre-	Post-
**MU Parameters**				
CSA of VL (cm^2^)	10.7 (4.6)	12.2 (4.3) *	12.8 (2.6)	12.5 (2.6) *
MT (cm)	1.7 (0.4)	1.9 (0.4) *	1.9 (0.2)	1.8 (0.3) *
FL (cm)	7.2 (0.5)	7.3 (0.8)	7.7 (0.7)	7.5 (0.7) *
PA (in°)	15.0 (3.9)	15.0 (4.1)	13.9 (3.2)	13.6 (3.0) *
CSA of LM left (cm^2^)	9.8 (2.7)	10.4 (2.3) *	10.7 (3.4)	11.2 (4.0)
CSA of LM right (cm^2^)	9.1 (2.1)	9.2 (2.6) *	10.3 (3.5)	10.3 (3.4)
**Functional Parameters**				
FTSST (in secs)	6.5 (2.1)	6.5 (2.6)	6.5 (0.4)	6.6 (0.7)
TUG (in secs)	4.7 (1.1)	4.4 (1.5)	4.6 (0.6)	4.6 (0.5)

Note: Values are presented as medians (IQR); * Wilcoxon Signed Rank test *p* < 0.05; (NMES—neuromuscular electrical stimulation group; CG—control group; MU muscle ultrasound; CSA—cross-sectional area; VL—vastus lateralis; MT—muscle thickness; FL—fiber length; PA—pennation angle; LM—lumbar multifidus; FTSST—Five Times Sit-to-Stand Test; TUG—Timed Up and Go test).

**Table 4 life-10-00184-t004:** Mean percentage changes and effect sizes in all the measured parameters.

	NMES *n* = 8	CG *n* = 8	*p* Value *	r
MU Parameters				
CSA of VL	11.3 (15.1)	−1.4 (4.0)	0.006	0.68
MT	6.9 (6.9)	−2.2 (4.7)	0.001	0.84
FL	2.1 (5.6)	−1.1 (4.2)	0.046	0.50
PA	4.0 (9.0)	−0.6 (2.9)	0.006	0.68
CSA of LM left	6.4 (8.2)	−0.6 (1.9)	0.036	0.53
CSA of LM right	6.5 (9.8)	−0.8 (2.3)	0.005	0.71
**Functional Parameters** #				
FTSST	−0.03 (23.0)	2.1 (11.7)	0.345	0.24
TUG	−10.9 (13.8)	2.6 (6.5)	0.046	0.50

Note: Values are presented as medians (IQR); * Mann–Whitney U test; r = effect size; # Negative values mean a positive effect; (NMES—neuromuscular electrical stimulation group; CG—control group).

**Table 5 life-10-00184-t005:** Schedule of combined NMES training.

Combined NMES Training	Week	Duration	Frequency and Pulse	Intensity	Pre-Load	Load
Quadriceps	1–8 w (3/w)	18′	75 Hz Rectangular biphasic waves (400 µs)	Maximum tolerable intensity corresponded to the individual’s pain threshold	5′ pedaling	40 contractions of 6.25 s (rise time 1.5 s, steady time 4 s, fall time 0.75 s)
Lumbar paraspinal muscles	1–8 w (3/w)	15′	50 Hz Symmetrical biphasic waves (300 µs)		5′ pedaling	45 contractions of 10 s (rise time 1 s, steady time 8 s, fall time 1 s)

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
