# Peer review of "Muscle Hypertrophy and Architectural Changes in Response to Eight-Week Neuromuscular Electrical Stimulation Training in Healthy Older People"

_life, 2020, doi:10.3390/life10090184_

Round 1
Reviewer 1 Report
In the manuscript “Muscle hypertrophy and architectural changes in response to eight-week neuromuscular electrical stimulation training in healthy older people” T. Jandova and coworkers evaluated changes of several parameters in lumbar multifidus (LM) and vastus lateralis (VL) muscles after a 8-week neuromuscular electrical stimulation (NMES) cycle in elderly people. The work, which includes also functional tests, is a well-conducted pilot study.
Although not completely novel in its approach (i.e., the use of NMES to induce hypertrophy of muscle mass), the work investigated several architectural parameters by the non-invasive method of muscle ultrasound. The authors found that, while control group showed progressive loss of muscle functionality, 8-week NMES training resulted in increased muscle thickness and cross-sectional area especially in VL muscles, suggesting NMES as a useful tool for combating age-related loss of muscle mass and function (i.e., sarcopenia).
The manuscript is extremely clear and well written. A comprehensive Discussion enriched the manuscript.
Minor points
1) Line 21: Replace “improve” with “improves”.
2) Line 74: “MVC” should be defined.
3) Line 99: “CG” should be defined.
Author Response
We would like to thank the referee for the positive comments and for spotting the few typing errors in the text.
Reviewer 2 Report
Muscle hypertrophy and architectural changes in response to eight-week neuromuscular electrical stimulation training in healthy older people
by T. Jandova et al.
is a well designed and implemented study in an area in which many studies are published.
Unfortunately, the authors do not report some key studies in Introduction and/or discussion. A list of mandatory citation follows:
- Mayr, W. Neuromuscular Electrical Stimulation for Mobility Support of Elderly. J. Transl. Myol. 2015, 25, 263–268, doi:10.4081/ejtm.2015.5605.
- Protasi, F. Mitochondria Association to Calcium Release Units is Controlled by Age and Muscle Activity. J. Transl. Myol. 2015, 25, 257–262, doi:10.4081/ejtm.2015.5604.
- Sarabon, N.; Löfler, S.; Hosszu, G.; Hofer, C. Mobility Test Protocols for the Elderly: A Methodological Note. J. Transl. Myol. 2015, 25, 253–256, doi:10.4081/ejtm.2015.5385.
- Cvecka, J.; Tirpakova, V.; Sedliak, M.; Kern, H.; Mayr, W.; Hamar, D. Physical Activity in Elderly. J. Transl. Myol. 2015, 25, 249–252, doi:10.4081/ejtm.2015.5280.
- Forcina, L.; Cosentino, M.; Musarò, A. Mechanisms Regulating Muscle Regeneration: Insights into the Interrelated and Time-Dependent Phases of Tissue Healing. Cells 2020, 9, 1297, doi:10.3390/cells9051297.
- Sajer, S.; Guardiero, G.S.; Scicchitano, B.M. Myokines in Home-Based Functional Electrical Stimulation-Induced Recovery of Skeletal Muscle in Elderly and Permanent Denervation. J. Transl. Myol. 2018, 28, 7905, doi:10.4081/ejtm.2018.7905.
- Scicchitano, B.M.; Sica, G.; Musarò, A. Stem Cells and Tissue Niche: Two Faces of the Same Coin of Muscle Regeneration. J. Transl. Myol. 2016, 26, 6125, doi:10.4081/ejtm.2016.6125.
- Barberi, L.; Scicchitano, B.M.; Musaro, A. Molecular and Cellular Mechanisms of Muscle Aging and Sarcopenia and Effects of Electrical Stimulation in Seniors. J. Transl. Myol. 2015, 25, 231–236, doi:10.4081/ejtm.2015.5227.
- Taylor, M.J.; Fornusek, C.; Ruys, A.J. Reporting for Duty: The duty cycle in Functional Electrical Stimulation research. Part I: Critical commentaries of the literature. J. Transl. Myol. 2018, 258 7732, doi:10.4081/ejtm.2018.7732.
- Taylor, M.J.; Fornusek, C.; Ruys, A.J. The duty cycle in Functional Electrical Stimulation research. Part II: Duty cycle multiplicity and domain reporting. J. Transl. Myol. 2018, 258, 7733, doi:10.4081/ejtm.2018.7733.
- Taylor, M.J.; Schils, S.; Ruys, A.J. Home FES: An Exploratory Review. J. Transl. Myol. 2019, 29, 8285, doi:10.4081/ejtm.2019.8285.
- Quittan, M.; Sochor, A.; Wiesinger, G.F.; Kollmitzer, J.; Sturm, B.; Pacher, R.; Mayr, W. Strength improvement of knee extensor muscles in patients with chronic heart failure by neuromuscular electrical stimulation. Organs 1999, 23, 432–435, doi:10.1046/j.1525-1594.1999.06372.x.
- Deley, G.; Denuziller, J.; Babault, N. Functional electrical stimulation: Cardiorespiratory adaptations and applications for training in paraplegia. Sports Med. 2015, 45, 71–82, doi:10.1007/s40279-014-0250-2.
- Braz, G.P.; Russold, M.F.; Fornusek, C.; Hamzaid, N.A.; Smith, R.M.; Davis, G.M. Cardiorespiratory and Muscle Metabolic Responses During Conventional Versus Motion Sensor-Assisted Strategies for Functional Electrical Stimulation Standing After Spinal Cord Injury. Organs 2015, 39, 855–862, doi:10.1111/aor.12619.
- Crevenna, R.; Wolzt, M.; Fialka-Moser, V.; Keilani, M.; Nuhr, M.; Paternostro-Sluga, T.; Pacher, R.; Mayr, W.; Quittan, M. Long-term transcutaneous neuromuscular electrical stimulation in patients with bipolar sensing implantable cardioverter defibrillators: A pilot safety study. Organs 2004, 28, 99–102, doi:10.1111/j.1525-1594.2004.40006.x.
- Coste, C.A.; Bergeron, V.; Berkelmans, R.; Martins, E.F.; Fornusek, C.; Jetsada, A.; Hunt, K.J.; Tong, R.; Triolo, R.; Wolf, P. Comparison of strategies and performance of functional electrical stimulation cycling in spinal cord injury pilots for competition in the first ever CYBATHLON. J. Transl. Myol. 2017, 27, 7219, doi:10.4081/ejtm.2017.7219.
Author Response
We would like to thank the referee for the positive comments and for suggesting additional references on NMES to be added to the manuscript. As the present is an experimental study and not a review on NMES, we condensed the list of references to those specifically on older individuals and also added a couple more published in journals other than J. Transl. Myol. as citing only this journal may have seemed biased.